# Effects of B on the Structure and Properties of Lead-Tin Bronze Alloy and the Mechanism of Strengthening and Toughening

**DOI:** 10.3390/ma14247806

**Published:** 2021-12-16

**Authors:** Xiaoyan Ren, Guowei Zhang, Hong Xu, Zhaojie Wang, Yijun Liu, Fenger Sun, Yuanyuan Kang, Mingjie Wang, Weize Lv, Zhi Yin

**Affiliations:** 1Department of Mechanical Engineering, Taiyuan Institute of Technology, Taiyuan 030008, China; renxiaoyan03@126.com; 2School of Materials Science and Engineering, North University of China, Taiyuan 030051, China; xuhong@nuc.edu.cn (H.X.); wzj1997vip@163.com (Z.W.); 155366264094@163.com (Y.L.); sunfengr@126.com (F.S.); kangyy469627632@163.com (Y.K.); 15513882577@163.com (M.W.); lwz-1st@163.com (W.L.); yinzhi1109@126.com (Z.Y.)

**Keywords:** lead–tin bronze, microstructure, grain refinement, mechanical properties, abrasion reduction performance

## Abstract

High lead–tin bronze is widely used in the selection of wear-resistant parts such as bearings, bearing bushes, aerospace pump rotors, turbines, and guide plates because of its excellent wear resistance, thermal conductivity, fatigue resistance, and strong load-bearing capacity. At present, high lead–tin bronze is used as a material for bimetal cylinders, which cannot meet the requirements of high-strength, anti-wear in actual working conditions under high temperature, high speed, and heavy load conditions, and is prone to de-cylinder, cylinder holding, copper sticking, etc. The reason for the failure of cylinder body parts is that the strength of copper alloy materials is insufficient, the proportion of lead in the structure is serious, and the wear resistance of the material is reduced. Therefore, it has important theoretical significance and application value to carry out research on the comprehensive properties of high-performance lead–tin bronze materials and reveal the strengthening and toughening mechanism. In this paper, The ZCuPb20Sn5 alloy is taken as the main research object, and the particle size, microstructure, mechanical properties, and friction of lead particles in ZCuPb20Sn5 alloy are systematically studied after single addition of B in ZCuPb20Sn5 alloy liquid. This paper takes ZCuPb20Sn5 alloy as the research object to study the effect of adding B on the morphology, microstructure, mechanical properties, and friction and wear properties of ZCuPb20Sn5 alloy lead particles, and discusses the strengthening and toughening mechanism of ZCuPb20Sn5 alloy under the action of B, and prepares a double high-performance lead–tin bronze alloy for metal cylinders. The main research results are as follows: The addition of B elements has an obvious refining effect on the α (Cu) equiaxed grains and lead particles in ZCuPb20Sn5 alloy. The average size of lead particles decreases from 30.0 µm to 24.8 µm as the B content increases from 0 wt.% to 0.1 wt.%. The reason for grain refinement is that B is easily concentrated at the grain boundary during the ZCuPb20Sn5 alloy solidification process, which affects the diffusion of solute atoms at the solidification interface, inhibits the grain growth, refines the grain, and hinders the sinking and homogenizes distribution between dendrites of lead; the tensile strength of the ZCuPb20Sn5 alloy improves. Relatively without B, when the addition of P is 0.1 wt.%, the tensile strength is the largest at 244.04 MPa, which enhances 13%; the maximum hardness gets 75.0 HB, which enhances 13.6%, as well as elongation get the maximum value at 17.2%. The main mechanism is that the addition of B forms a high melting point submicron Ni_4_B_3_ phase in the lead–tin bronze alloy. The Ni_4_B_3_ phase is dispersed in the matrix and strengthens the matrix. With the increase in B content (more than 0.1 wt.%), the Ni_4_B_3_ phase changes from sub-micron degree granular to micron degree block-like, and some defects such as shrinkage and porosity appear in the structure, resulting in a decrease in mechanical properties.

## 1. Introduction

The room temperature mechanical properties and processing properties of copper alloys are improved [1] on adding trace rare earths to copper and copper alloys. However, the effect of deterioration and refinement is still immature, and the refinement system is not perfect. Therefore, further research on the mechanism of copper alloys is needed. At the same time, with the need for the production of high-strength and high-conductivity copper alloys and the recycling of waste copper, the role of boron in copper alloys [2,3] has attracted more and more attention. Studies have found that trace boron has the effect of refining grains in copper alloys. At the same time, it is concluded that the optimal addition amount of B is 0.01–0.1 wt.%.

When Sun Liping studied the effects of Ce and B on the structure and properties of HSn70-1 alloy, he found when the H70, HAl77-2, and HAl77-2As brass contain 0.049%, 0.061%, and 0.091% (mass fraction) of B, respectively, the grains are significantly refined and the tensile strength increases by 40–50 MPa. The yield strength is increased by 40–60 MPa, and the elongation and reduction of area are slightly reduced, but the degree of reduction is relatively low, and it does not affect the good plasticity and workability of the alloy [4,5]. After adding boron to the 70Cu-30Ni alloy, the strength of the alloy can be increased. Similarly, the plasticity is slightly reduced, but the corrosion resistance is improved to a certain extent, and the wear resistance is significantly improved [6]. As the content of B increases, the refinement effect increases. The B content is between 0.008% and 0.015%, and the grain refinement and deterioration effect is the best when it is modified and poured after 10 min of heat preservation [7]. The addition of B can suppress the grain-coarsening phenomenon that occurs in the material for a longer time or at a higher temperature [8]. When the addition amount of B is in the range of 0~0.15 wt.%, with the increase in boron content, the dendrites in the CuNiMnFe alloy structure are gradually refined, and the distance between the secondary crystal arms is gradually reduced. When the addition amount of boron is 0.105 wt.%, the lath eutectic β phase basically disappears, the distance between the secondary arms of the dendrite in the alloy structure is the smallest, the as-cast hardness of the alloy reaches its peak, and the tensile strength of the alloy can reach 1130 MPa [9].

Boron is an important strategic substance in the 21st century. It has active chemical properties at high temperatures and is often used as a degassing agent. The solid solubility of boron in Cu at room temperature is 0.06%, and its limit is 0.53%. Boron is an element with a small diameter that can be dissolved in copper both in an interstitial manner and in a replacement manner, and B and Cu cannot form any compound [10]. The main mechanism is that after adding B and P, P and B can also be concentrated in intragranular defects such as dislocations, occupying a large number of vacancies or interstitial positions, reducing the stacking fault energy of the alloy, and producing solid solution strengthening. At the same time, phosphorus and boron inhibit the grain boundary diffusion and are also beneficial to inhibit the grain boundary slip and crack initiation during creep. This delays the recovery of the alloy, inhibits the formation of recrystallized cores and the growth of crystals. Therefore, it is possible to refine the grains and purify the material, so that the alloy can obtain excellent high-strength comprehensive properties.

Because there have always been different opinions on the role of phosphorus in alloys, how to reasonably control the amount of phosphorus added to the alloy has become a difficult problem. On the basis of the existing research results, when the P element addition amount is controlled to 0.1 wt.%, it becomes more and more important to improve the mechanical properties of the lead–tin bronze alloy by adding B element instead of high phosphorus.

Since there have always been different opinions on the role of phosphorus in alloys, how to reasonably control the amount of phosphorus added to the alloy has therefore become a problem. In this paper, based on the existing research results, the addition of P element is controlled to 0.1 wt.%, and B is added instead of high phosphorus to improve the mechanical properties of the lead–tin bronze alloy.

## 2. Experimental Procedure

### 2.1. Preparation of As-Cast ZCuPb20Sn5 Alloy

ZCuPb20Sn5-yB alloy castings are smelted in a well-type resistance furnace and a 16# graphite crucible. In ZCuPb20Sn5-yB alloy, y = (0, 0.01, 0.05, 0.1 and 0.2). The smelting furnace model is SG2-12-13, the rated voltage is 380 V, and the rated temperature is 1300 °C. Before smelting, raw materials, casting molds, and graphite rods for stirring are put in a 300 °C holding furnace in advance to preheat. During the smelting process, the temperature of the smelting furnace is adjusted to 900, 1100, and 1250 °C in sequence, and the furnace temperature is slowly increased. Electrolytic pure copper is the main raw material. Oxidation is inevitable during alloy smelting. In the smelting of lead–tin bronze, in order to prevent oxidation problems, the deoxidizer used in the experiment is a phosphor-copper alloy with a P content of 13.5 wt.%. Due to the longer smelting process, the addition of alloys, and the number of steps, the deoxidation is carried out in two stages. When the first copper block is completely melted, 2/3 of the phosphorous copper is added for preliminary deoxidation; after the addition of alloying elements is completed, the remaining 1/3 of the phosphorous copper is added.

### 2.2. Test and Analysis

The microstructures of the as-cast ZCuPb20Sn5-0.1P-yB were prepared, and then samples were taken to study the effect of B on the structure change of ZCuPb20Sn5 alloy. The size of the metallographic analysis sample is basically 15 mm × 15 mm × 15 mm. After the sample was corroded by ammonia, hydrogen peroxide, and water in a ratio of 1:1:3, the morphology was observed with the AXIO Scope.A1 metallographic microscope. And, using the grain evaluation software that comes with the device, the size and morphology of the lead particles in the ZCuPb20Sn5 alloy structure were evaluated. Analyze the size and morphological distribution of lead particles in the same field of view to determine whether there was segregation of lead particles in the organization. The morphology of the copper alloy was observed on Hitachi’s new generation of thermal field emission scanning electron microscope SU5000, and combined with its built-in energy spectrometer function to perform energy spectrum analysis on the composition of the alloy structure and the composition of the precipitated phase, and at the same time stretched the fracture Scanning and data were also done using this device. A D/max-rB X-ray diffractometer of Rigaku (RIGAKU) was used to analyze the phase composition of the sample. The scanning speed was 6 deg/min, and the scanning angle was 10~90°. The Setaram Labsys synchronous thermal analyzer was used to detect and analyze the DSC experimental data, and measure the heat flow under different temperature conditions. The temperature measurement range is 20~1400 °C, the heating rate is 5 °C/min, and the cooling rate is 10 °C/min. The copper alloy samples tested were of the size and shape of millet grains. The graph ICP-AES chemical element analyzer was used to detect the fluctuation of the chemical composition after each experiment to ensure the accuracy of the experimental data. Using the first-principles method based on density functional theory, the stability of the alloy was judged by the calculation results of the enthalpy of formation and binding energy, and the stability, hardness, and ductility of the second phase particles were simulated and predicted

The analysis of the tensile strength of ZCuPb20Sn5 alloy castings at room temperature was completed on the Instron8801 static mechanical testing machine. Hardness testing was done in accordance with the national standard of “Standard Hardness Block Calibration” GB/T 231.1-2018 “Metallic Material Brinell Hardness Test Part 1: Test Method”, using the HB-3000C electronic Brinell hardness tester to complete.

## 3. Results and Discussion

### 3.1. Microstructures

#### 3.1.1. The Change of Lead Particle Morphology in ZCuPb20Sn5 Alloy by Adding B

The microstructure of ZCuPb20Sn5-0.1P alloy samples with different B content is shown in Figure 1.

Figure 1a–e are metallographic photos added 0 wt.%, 0.01 wt.%, 0.05 wt.%, 0.1 wt.%, 0.2 wt.% B, respectively, to ZCuPb20Sn5-0.1P alloy. It can be seen that the lead particles in the alloy structure tend to be refined after adding boron from the figure. Specific changes can be analyzed in conjunction with Figure 2.

Figure 2a–e is respectively shows the change in lead particles and distribution map of lead particles in ZCuPb20Sn5-0.1P alloy after adding different content of boron (0 wt.%, 0.01 wt.%, 0.05 wt.%, 0.1 wt.%, and 0.2 wt.%). From the data in Figure 2, it can be seen that the amount of B added is between 0.01% wt.% and 0.2% wt.%, and the average size of lead particles is less than 250 μm. Figure 2a shows that when B is not added, there are 22 lead particles in the range of 120–250 μm and it is reduced to 14 when 0.01 wt.% B is added, as shown in Figure 2b. When the content of B is added to 0.1 wt.%, as shown in Figure 2d, the lead particles in the size range of 120–250 μm are reduced to 9 pieces. Compared with other types of B content, 0.1 wt.% of lead is added. In the case of particles, the lead particles in the two ranges of 0–30 μm and 15–30 μm are the most. With the increase in B content, the lead particles in the size range of 60–120 μm gradually decrease, from 192 without B added to 98 when B 0.1 wt.% is added.

It shows that the addition of B can refine the lead particles, but when the content of B increases to 0.2 wt.%, as shown in Figure 2e, the size of the lead particles increases, and the shape gradually changes from spherical to irregular.

From the change in average diameter of the lead particles (Table 1), it can be seen that after adding B to the alloy liquid ZCuPb20Sn5, the lead particles are obviously refined. With the addition of B, the average diameter of lead particles in the alloy is refined from 30.0 µm when B is not added to 24.8 µm when B is added 0.1%, which is 17.3%. However, as the B content continues to increase, the lead particles have a tendency to grow again.

It can be clearly seen from Figure 3 that the addition of boron can refine the lead particles and make the large lead particles smaller and more uniform. Among them, the small particles (lead particles in the range of ≦30 μm) increase with the increase of B content. As the B content increases, the large particles (lead particles in the range of 30 μm) decrease with the increase of B content. But when the content of B exceeds 0.2 wt.%, although the small particles increase, the large particles in the range of 30~120 also increase. From the perspective of the change form of lead particles, the addition of element B cannot be greater than 0.2 wt.%.

The influence of the size and number of lead particles on the organization and performance will be described in detail in the following organization performance analysis section. However, the size of the lead particles is too large and unevenly distributed, which will cause the organization to deteriorate. Therefore, reduce the size of the lead particles at the same time to make it evenly distributed is one of the purposes of its alloying.

At the same time, when B is not added, there are lead particles in the range of 60~500 μm. After adding B, the large particles are mainly concentrated in the range of 120~250 μm, and the number is small. After adding B 0.01 wt.%, the number of lead particles in this range is 14. With the increase of B content, the lead particles tend to be refined, but the effect will be weakened after adding more.

The main reasons for the changes in lead particles:When B is added to the copper liquid, the solidification range of the copper alloy is shortened, and the time for the lead particles to sink and gather, so the lead particles become smaller and evenly distributed.The addition of B refines the crystal grains and prevents the sinking of lead.The addition of B reflects the harmful impurities in the liquid, removes harmful gases such as oxygen and hydrogen, and makes the lead particles pure, fine, and evenly distributed.As the content of B continues to increase (more than 0.1%), the form of the second phase with high melting point will change from granular to massive, or even large lumps, which will reduce the fluidity of the alloy liquid and cause the lead particles to gather and grow up.

#### 3.1.2. Change of α Structure Morphology in ZCuPb20Sn5-0.1P-yB Alloy Structure


*(1) Macro Organizational Changes*


After sampling, polishing, and etching the castings after casting, observe their macroscopic appearance, as shown in Figure 4.

Figure 4a–d is photos of the macrostructure of the alloy with B (0.01 wt.%~0.2 wt.%). It can be seen from the figure that the addition of B refines the α grains. When 0.1 wt.% of B is added, the α grains are the smallest, and the content of B continues to increase, the grains begin to grow.


*(2) Microstructure*


After corrosion of ZCuPb20Sn5 alloy samples with different B content (0 wt.%, 0.01 wt.%, 0.05 wt.%, 0.1 wt.% and 0.2 wt.%), they were observed under a scanning electron microscope to obtain the microstructure of the sample, as shown in Figure 5.

The content of trace element B is different, and the dendrite morphology is different in the area. From Figure 5a–e, it can be seen that the α dendrites are distributed in equiaxed crystals, and the crystal axis is refined with the increase of B content. However, when the content of B is greater than 0.1 wt.%, as shown in Figure 5e, the dendritic structure changes, from equiaxed crystals to disordered chrysanthemums, and the crystal arms become thicker. In the four different solutions when the B content is less than 0.2 wt.%, the thickest dendrite arm is about 20 μm, and the average length is between 20–60 μm. With the increase of B content, the number of dendrite arms in the area increases, and the dendrite arms become finer. When the B content is increased to 0.1 wt.%, the dendrite size in the area is on average between 10–15 μm. When the content of B increased to 0.2 wt.%, the organization changed, and the original dendrites began to distribute in the form of chrysanthemums.

#### 3.1.3. Analyze the Reasons for the Refinement of Alpha Phase and Lead Particles by B

Figure 6 shows the relationship between heat flow and sample temperature during the cooling process of ZCuPb20Sn5 and ZCuPb20Sn5B0.1 alloys. The black line represents the curve of ZCuPb20Sn5P0.1 alloy, and the red represents the curve of ZCuPb20Sn5B0.1.

It can be seen from Figure 6 that the overall curve has three obvious exothermic peaks during the cooling process. The ZCuPb20Sn5B0.1 alloy has four exothermic peaks. It corresponds to the three peaks of the ZCuPb20Sn5 alloy. The first exothermic peak of the ZCuPb20Sn5 alloy is at (968.83 °C, −14.17 w·g^−1^), while the second exothermic peak of the ZCuPb20Sn5B0.1 alloy is at (985.47 °C, −4.66 w·g^−1^), where are the precipitated phases of the copper matrix. The temperature of the copper matrix phase is reduced by 17 °C in the first place. The temperature of the second eutectoid (α + δ + Ni_3_P) is reduced by 11 °C. The solidification temperature range of the copper phase is shortened, and the large lead particles are distributed evenly among the dendrites before they can gather and grow up. With the addition of B, the intragranular structure is refined, and the lead particles gradually change from large dots to small spheres, which improves the degree of lead segregation.

The second peak of the ZCuPb20Sn5 alloy and the third exothermic peak of the ZCuPb20Sn5B0.1 alloy are at (904.54 °C, −14.22 w·g^−1^) and (893.35 °C, −4.96 w·g^−1^), respectively. According to the above phase diagram, it is judged that this place is the precipitation point of (α + δ + Ni_3_P) eutectoid. The third peak of the ZCuPb20Sn5 alloy and the fourth exothermic peak of the ZCuPb20Sn5B0.1 alloy are at (323.48 °C, −0.64 w·g^−1^) and (322.89 °C, −0.29 w·g^−1^), respectively. This is the peak point of the precipitation phase of lead particles. The melting point of lead is 328 °C, and the crystallization point of lead is reduced by 4 to 5 °C. After adding boron, as shown by the arrows in Figure 6, the size of the lead particles is different, resulting in a slightly different precipitation temperature of the lead particles—the lead particles become smaller, and the peak intensity gradually weakens.

The first exothermic peak of the ZCuPb20Sn5B0.1 alloy has a small peak at about 1025 °C. According to the previous Cu-Ni phase diagram, it can be seen that the melting points of Ni_2_B and Ni_3_B are 1125 °C and 1156 °C, respectively. Since the melting point of Ni_4_B_3_ is 1018 °C, this point is the precipitation point of Ni_4_B_3_ phase, which is consistent with the results of XRD data. During the solidification of the alloy material, the Ni_4_B_3_ phase is first precipitated at 1025 °C. Since the number of this phase is extremely small, the peak value is particularly small. Subsequently, this phase is formed before the solidification of copper, as a solute component, which is dissolved in the copper alloy liquid. As the solidification of the copper alloy is precipitated at 968~985 °C, the peak value is relatively large, and the copper matrix phase begins to precipitate. Then, when the temperature was lowered to 898~904 °C, the eutectoid (α + δ + Ni_3_P) precipitated, and the precipitation temperature of the lead particles was 323 °C.

The Ni_4_B_3_ phase is formed before the solidification of copper, and as the copper alloy solidifies and precipitates, it is repelled by the copper grains and concentrated at the grain boundary, which directly affects the diffusion of solute atoms at the solidification interface, the growth curvature of the interface, and the re-nucleation occurrence and so on, thereby inhibiting the growth of grains and achieving the purpose of grain refinement. The main refinement effect is to limit the growth mechanism.

#### 3.1.4. Microstructure and Phase Structure of the ZCuPb20Sn5-yB Alloy


*(1) XRD Analysis*


In order to judge the phases in the structure more accurately, further analysis of the alloy is needed. Figure 7 is the XRD data diagram of the alloy after adding B, and Figure 7a is the XRD data graph of the ZCuPb20Sn5-0.1B alloy, Figure 7b is the XRD data graph of alloy ZCuPb20Sn5-0.2B.

It can be seen from Figure 9 that a series of main peaks clearly appear in the XRD spectrum, which are consistent with the face-centered cubic (FCC) solid solution α phase (Cu-based) and represent the diffraction peaks of the copper matrix. By analyzing the XRD data, it can be seen that when the addition amount of B is 0.1 wt.%, the second phase of P is mainly the Ni_3_P phase, which appears at 36.418° (031) and the peak value is relatively strong. The second phase of B is mainly NiB_12_ phase; this phase is less, and this phase has a coherent relationship with the copper matrix phase. The content of phosphorus and nickel is different, and the atomic ratio of nickel and boron is different. In the XRD diffraction pattern of alloy ZCuPb20Sn5-0.2B, the NiB_12_ phase is basically not detected, but the diffraction peak of Ni_4_B_3_ phase appears.


*(2) Organizational Structure*


The ZCuPb20Sn5-yB alloy was analyzed by SEM and EDS, and the crystal structure of the mesophase that may be formed by different B in the alloy ZCuPb20Sn5 was analyzed, as shown in Figure 8 and Figure 9.

Figure 8 and Figure 9 are the phase structure diagrams of alloy ZCuPb20Sn5-yB. Figure 8a–c are respectively SEM images of the ZCuPb20Sn5-yB alloy (y = 0 wt.%, 0.01 wt.% and 0.05 wt.%). Figure 8d–j are the circles A, B, C, D, E, F, and G in the figure. The EDS data graph is pointed by the arrow. Figure 9a,b,g are the SEM images of the ZCuPb20Sn5-0.1P-yB alloy (y = 0.1 wt.% and 0.2 wt.%), respectively. Figure 8c–f,h–j is the arrows in the circles of A, B, C, D, and E in the figure graph of EDS data.

It can be seen from the figure that when the content of B added is less than 0.1 wt.%, the second phase Cu_3_P is not found in the structure, except for the matrix α phase, Pb phase, and (α + δ) phase. From the SEM of Figure 8, the small black lumps indicated by A, D, and E, and the corresponding EDS data Figure 8d,g,h confirm that it may be the CuNi_2_P phase, and in this range, 10 similar particles are selected for measurement and the average value is obtained, and it is concluded that the phase is close to the CuNi_2_P phase. White irregular spherical particles are Pb phase particles, and off-white block. From the SEM of Figure 8b,c indicate the off-white block and island block structure, and the corresponding EDS data in Figure 8e,f, it can be confirmed that this phase is a δ phase.

When the addition amount of B is greater than or equal to 0.1 wt.%, as shown in Figure 9, the phase structure in the ZCuPb20Sn5-0.1P-yB alloy structure changes, and the δ phase in the ZCuPb20Sn5-0.1P-yB alloy structure decreases. There is also a black–gray flaky boron phase in the structure. After EDS composition analysis, the ZCuPb20Sn5-0.1B alloy is shown at A and C in Figure 9 SEM, corresponding to the EDS images (a) and (c). It is known that the lamella structure is a phase of nickel and boron. According to the ratio of atomic number, it can be preliminarily judged that this item is a Ni_2_B phase. From the positions B and D in Figure 8 SEM, corresponding to the EDS diagrams (b) and (d), it can be seen that the small black dots are mainly the solid solution phase of boron and copper, because according to the phase diagram, it can be seen that boron and copper has no binary alloy phase, and the boron phase is mainly the NiB_12_ phase. When the content of boron increases, the boron–nickel phase changes, as shown in the EDS picture 9 (h) corresponding to E in the SEM of Figure 9, the block shape becomes bar shape, and the structure becomes Ni_3_B.


*(3) Crystal Structure Analysis*


From the above analysis, it can be known that the Ni_4_B_3_ phase may be formed in the ZCuPb20Sn5-yB alloy. The first-principles simulation software is used to simulate and calculate the Ni_4_B_3_ phase in the ZCuPb20Sn5-yB alloy; using the density functional theory Cambridge Serial Total Energy Package (Castep) [11] package module, periodic boundary condition, the crystal wave function is expanded by the plane wave group [12]. The plane wave cut-off energy Ecut is 540.0 eV, the number of K-point grids for the Ni_3_P phase is 6 × 6 × 12, and the Cu3P phase is 10 × 10 × 10. The plane wave truncation energy is 500 eV, and the number of K-point grids of the Ni_4_B_3_ phase is 10 × 12 × 8. The Broyden Flecher Goldfarb Shanno (BFGS) method [13] is used for geometric optimization to obtain their local stable structures. The simulated Ni_4_B_3_ phase structure model is shown in Figure 10, and the simulation data results are shown in Table 2.

From the data in Table 2, we can see that the Ni_4_B_3_ phase has a triclinic crystal structure. The density is close to that of the Cu_3_P phase, and the volume is smaller than that of Cu_3_P and Ni_3_P. In addition, after calculation, the absolute values of the formation enthalpy and binding energy of Ni_4_B_3_ phase are greater than Cu_3_P and Ni_3_P. With the larger absolute value, the easier it is to form and the more stable after formation. Therefore, Ni_4_B_3_ is easier to form than Cu_3_P and Ni_3_P, and more stable after formation.

The addition of B has an obvious refinement effect on the structure of ZCuPb20Sn5 alloy, which can refine the α(Cu) equiaxed crystal and reduce the lead particles. When B is not added, the average diameter of lead particles is 30.0 µm, and when 0.1 wt.% B is added, it is 24.8 µm, which is 17.3% thinner. The reason for the grain refinement is that when the ZCuPb20Sn5 alloy is solidified, boron is easily enriched at the grain boundary, which affects the diffusion of solute atoms at the solidification interface, inhibits the growth of grains, refines the grains, prevents the sinking of lead, and distributes it evenly between the dendrites. When the amount of B added is 0.1 wt., the Ni_4_B_3_ phase is formed in the alloy.

### 3.2. The Effect of B on the Mechanical Properties of Lead–Tin Bronze Alloys, and the Strengthening and Toughening Mechanism

#### 3.2.1. Hardness Analysis

It can be seen from the bar (graph 11a) that the hardness of the ZCuPb20Sn5 alloy increases with the increase in boron. From not adding B to 0.01 wt.%, the hardness increased 1 HB. In the process of increasing from B 0.01 wt.% to B 0.05 wt.%, the hardness increased 6.56 HB, and the increase rate was 9.95 wt.%. In the process of increasing from B 0.05 wt.% to B 0.1 wt.%, the hardness increased by about 1.5 HB, and the increase was not large. When the B content is 0.1 wt.%, the hardness is 75 HB. When the B content is 0.2 wt.%, the hardness value drops to 74.2 HB. Judging from the hardness values of several groups of different contents of B, when 0.1 wt.% of B is added, the hardness value is the largest. However, it can be seen from the fitting curve of Figure 11b that the maximum hardness is between B 0.1 wt.% and B 0.2 wt.%.

The fitting curve is shown in Figure 11b. It can be seen that with the increase of B content, the hardness of ZCuPb20Sn5 alloy shows a parabolic increase and decrease trend. The law of increase and decrease satisfies the Formula (1), and every point in the experimental data is basically on the curve.
y_B_ = −580.13x_B_^2^ + 155.97x_B_ + 66.073(1)
where, y is the hardness of ZCuPb20Sn5 alloy, and the unit is HB. x_B_ is the content of B added, and the unit is: wt.%.

After analyzing the hardness of the above ZCuPb20Sn5-0.1P-yB alloy material, it can be found that the addition of B can increase the hardness of the material, and the increase law satisfies the change law of the formula y_B_ = −580.13 × B2 + 155.97 × B + 66.073. With the increase in B content, the hardness shows a trend of increasing first and then decreasing, with little change. Consistent with the change rule of tensile strength, when the addition amount of B is 0.1 wt.%, it reaches the maximum value of 75.0 HB. Compared with the absence of B, the hardness increased by 9 HB, an increase of 13.6%. The main reason for the increase in hardness is that the addition of B can generate second phase particles. From the ZCuPb20Sn5 alloy structure, it can be seen that the addition of B produces second phase particles, which leads to an increase in the hardness of the alloy material. But the content of this phase is very small, so the hardness change is not big, basically within the error range.

#### 3.2.2. Analysis of Tensile Strength Data

After sampling and measuring the cast ZCuPb20Sn5-yB alloy sample, the as-cast mechanical properties of different schemes at room temperature are obtained. According to the obtained mechanical property data, the columnar distribution diagram of the tensile strength of different elements is drawn, as shown in Table 3. According to the tensile strength of the alloy, Equation and curve in Figure 12b are fitted.

Figure 12a shows the change trend graph of the tensile strength of ZCuPb20Sn5 alloy with different element content. It can be seen from the bar graph that the tensile strength of the ZCuPb20Sn5 alloy increases with the increase in the amount of boron added. From no addition of B to 0.01 wt.%, the tensile strength increased by 3 MPa, an increase of 1.42% from 0.01 wt.% B to 0.05 wt.% B. In the process of increasing from 0.05 wt.% B to 0.1 wt.% B, the tensile strength increased by 20 MPa, with an increase of 9.09%; it can be seen that the phosphorus addition amount is less than or equal to 0.1 wt.% B. With the increase in boron content, the tensile strength increases, and the increase is not large; the maximum increase of the tensile strength is 9.09%. According to the fitting curve, it can be seen that the peak value is reached when the amount of B is 0.1 wt.%, and the tensile strength is 244 MPa.
y_B_ = −1071.7x_B_^2^ + 357.42x_B_ + 212.89(2)

Among them, y represents the content of B, the unit is wt.%; y_B_ is the corresponding tensile strength, the unit is MPa.

The addition of boron can combine with some elements in the ZCuPb20Sn5 alloy to produce high-melting-point boride, which serves as the nucleation matrix of the crystal grains during the solidification process to promote the nucleation rate, refine the structure, and improve performance. When B is added to ZCuPb20Sn5 alloy, the high melting point boride formed is mainly a nickel–boron phase. The addition of boron has an impact on Pb, S, P, etc., from the original distribution in grain boundaries and dendrites to uniform distribution in the crystal. The boride can completely spheroidize Cu_2_O and Cu_2_S or even disappear completely, and the inclusions can be precipitated from the solid solution state through the boride. In addition, in terms of lead segregation, high melting point borides can be distributed on the copper matrix, and α-phase dendrites are precipitated and refined in advance, which inhibits the enrichment of lead, reduces the segregation of lead, and improves the mechanical properties.

#### 3.2.3. Analysis of Elongation Data

Figure 13 is a histogram of the elongation distribution of ZCuPb20Sn5 alloy with different boron content; the elongation does not change significantly.

It can be seen from the histogram13 that the elongation of ZCuPb20Sn5 alloy shows a trend of first increasing and then decreasing with the increase of boron, but the increase and decrease are relatively small. The elongation is basically maintained between 16% and 17%, which is relatively stable. It shows that the addition of boron has little effect on the elongation of the alloy. However, the addition of boron can improve the mechanical properties, and an appropriate amount of boron can replace part of the role of phosphorus, and at the same time offset the hot embrittlement caused by high phosphorus.

The smaller the crystal grains, the more uniform the deformation—therefore, the smaller the stress concentration generated, which helps to improve the plasticity of the material. The addition of B can refine the structure of the ZCuPb20Sn5 alloy, so the plasticity is improved. At the same time, the smaller and finer the crystal grains, the greater the number of crystal grains in a certain volume. Therefore, when the amount of deformation is the same, the deformation in the crystal grains is dispersed in more crystal grains, so that the deformation in each crystal grain becomes smaller and uniform, and the excessive stress concentration phenomenon is eliminated. Moreover, as the crystal grains become smaller, the grain boundaries become more and more tortuous, thereby preventing the cracks from continuing to propagate and making the cracks undergo greater plastic deformation before fracture. In this way, metal materials exhibit higher plasticity and toughness. However, because the second phase particles are generally hard, their size is too large, which easily causes brittle fracture of the material during the deformation process, thereby reducing its elongation.

#### 3.2.4. Fracture Morphology Analysis

Figure 14 is the fracture scan photograph of ZCuPb20Sn5-yB alloy material.

It can be seen from Figure 14 that after adding B, there are obvious dimples at the fracture. Therefore, the fracture mode of the two materials is plastic fracture. Moreover, the addition amount is different, the size and shape of the dimples are different. It is known from the previous results that the elongation of ZCuPb20Sn5 alloy begins to decrease after adding boron over 0.05 wt.%. It can be seen from Figure 14 ZCuPb20Sn5-0.1P-yB alloy fracture morphology that after adding B0.1 wt.%, the size of the dimples in the tissue is not large, but the dimples are deep and relatively uniform. However, after adding 0.2 wt.% B, it is obvious that black large particles exist at the fracture dimples in the alloy structure, indicating that the inclusions or the second phase are not only the holes formed around the inclusions or precipitates when the material undergoes plastic deformation. The resulting microporous fracture, and with the growth of the second phase and inclusions, prevents dislocation slippage and creates holes, which leads to slippage separation fractures of the material. Therefore, when the B content is increased to 0.2 wt.%, the elongation decreases slightly.

Due to the addition of element B, the grain size of the ZCuPb20Sn5 alloy structure becomes smaller and the stress concentration is weakened, thereby improving the tensile strength of the alloy. At the same time, after the addition of B, the second phase hard particles generated are dispersed in the matrix grains, on the grain boundaries or aggregated into agglomerates, which hinder the dislocation during the solidification process and achieve the strengthening effect. Therefore, the strength of the ZCuPb20Sn5 alloy material is improved.

#### 3.2.5. Strengthening Analysis of the Second Phase Ni_4_B_3_ Particles

In the first principles, the CASTEP and Forcite Plus modules are used to simulate the second phase Ni_4_B_3_ appearing in the alloy ZCuPb20Sn5-yB, and the elastic constants of the particles in this phase are obtained in Table 4, and according to the relevant formula calculate the bulk modulus (B), shear modulus (G), Young’s modulus (E), and Poisson’s ratio (µ) of the Ni_4_B_3_ phase, as shown in Table 3.

From the data in Table 4, it can also be seen that the Young’s modulus and shear modulus of Ni_4_B_3_ are much greater than those of Ni_3_P. The Ni_4_B_3_ phase C44 is also larger than the Ni_3_P phase. Therefore, it can be judged that the Ni_4_B_3_ phase particles are hard particles with a hardness greater than that of Cu_3_P.

By calculating the G/B (Pugh ratio) value of Ni_4_B_3_ phase:
Pugh ratio of Ni_4_B_3_: G_V_/B_V_ = 100.22/225.45 = 0.445G_R_/B_R_ = 92.42/211.68 = 0.437G_H_/B_H_ = 96.32/218.57 = 0.441

The calculation results show that the G/B of the Ni_4_B_3_ phase is less than 0.5, but close to 0.5, indicating that this phase material is a ductile phase, but ductility is not high.

## 4. Conclusions

By adding different alloying elements to study the structure and properties of ZCuPb20Sn5 alloy, the following conclusions are drawn:

(1) The addition of B has an obvious refinement effect on the structure of ZCuPb20Sn5 alloy, which can refine the (Cu) equiaxed crystal and reduce the lead particles. When B is not added, the average diameter of the lead particles is 30.0 µm, and 0.1 wt.% B is added. It is 24.8 µm, which is 17.3% thinner. The reason for the grain refinement is that when the ZCuPb20Sn5 alloy is solidified, boron is easily enriched at the grain boundary, which affects the diffusion of solute atoms at the solidification interface, inhibits the growth of grains, refines the grains, prevents the sinking of lead, and distributes it evenly between the dendrites. When the amount of B added is 0.1 wt.%, Ni_4_B_3_ phase is formed in the alloy.

(2) The tensile strength of ZCuPb20Sn5 alloy material increases first and then decreases with the increase of B content. The formula y_B_ = −1071.7x_B_ + 357.42x_B_ + 212.89 can be used to describe the change law. The increase in material hardness and elongation is not obvious. When the content of B is 0.1 wt.%, the tensile strength reaches the maximum. Compared with the absence of B, the tensile strength increased by 28 MPa, an increase of 13%; the maximum hardness was 75.0 HB, an increase of 9 HB, the increase was 13.6%, and the elongation was the maximum 17.2%.

(3) The main reason for the increase in the strength of the ZCuPb20Sn5 alloy is that the addition of B can generate high melting point Ni_4_B_3_ particles, resulting in dispersion strengthening. At the same time, the addition of B can refine the grains, produce fine-grain strengthening, and increase the strength of the alloy. As the B content continues to increase, the resulting high-melting-point second phase changes from granular to lumpy, or even large-scale lumps, which reduces the fluidity of the alloy liquid and causes defects such as shrinkage and porosity in the structure, resulting in a decrease in mechanical properties.

## Figures and Tables

**Figure 1 materials-14-07806-f001:**
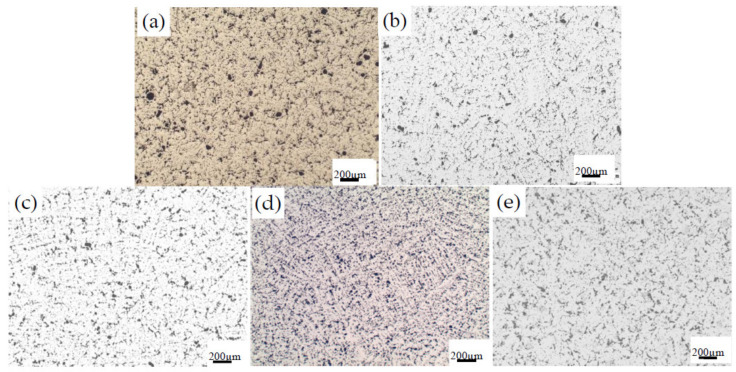
The microstructure of ZCuPb20Sn5-0.1P alloy with different content of B: (**a**) B 0 wt.%, (**b**) B 0.01 wt.%, (**c**) B 0.05 wt.%, (**d**) B 0.1 wt.%, (**e**) B 0.2 wt.%.

**Figure 2 materials-14-07806-f002:**
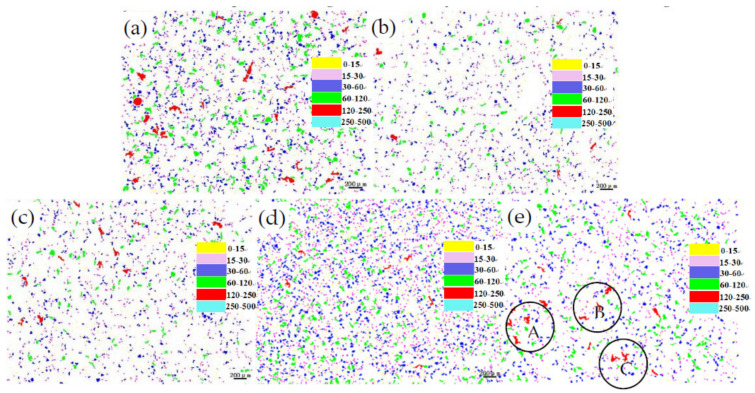
Changes of lead particle size distribution in ZCuPb20Sn5 alloy with different P contents: (**a**) B 0 wt.%, (**b**) B 0.01 wt.%, (**c**) B 0.05 wt.%, (**d**) B 0.1 wt.%, (**e**) B 0.2 wt.%.

**Figure 3 materials-14-07806-f003:**
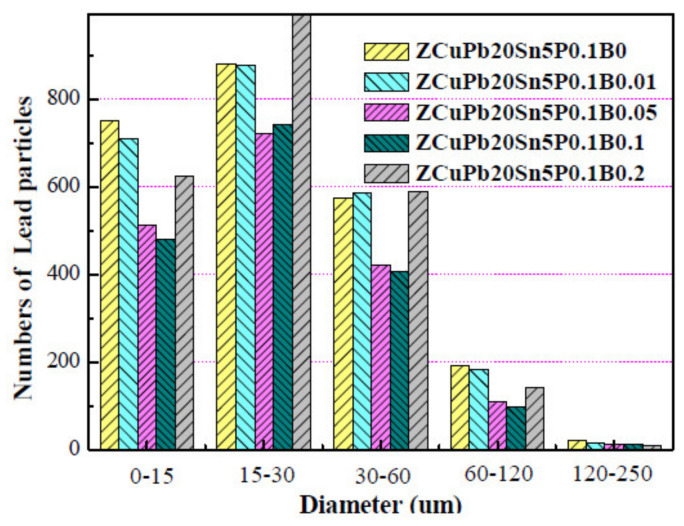
Lead particle number distribution map of ZCuPb20Sn5-yB alloy.

**Figure 4 materials-14-07806-f004:**
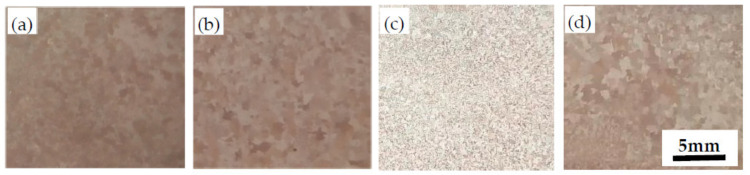
Macrostructure of the ZCuPb20Sn5Ni3-0.1P-yB alloy. (**a**) B = 0.0 wt.%; (**b**) B = 0.05 wt.%; (**c**) B = 0.1 wt.%; (**d**) B = 0.2 wt.%.

**Figure 5 materials-14-07806-f005:**
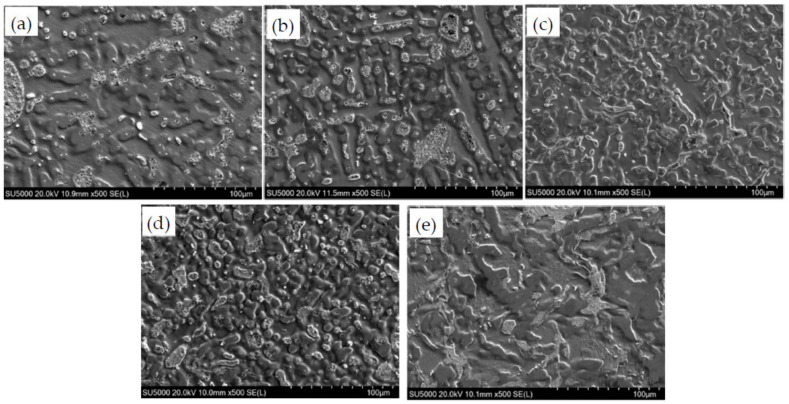
BSE microstructure of the ZCuPb20Sn5-0.1P-yB-zY alloy at 500 times. (**a**) B = 0 wt.%; (**b**) B = 0.01 wt.%; (**c**) B = 0.05 wt.%; (**d**) B = 0.1 wt.%; (**e**) B = 0.2 wt.%.

**Figure 6 materials-14-07806-f006:**
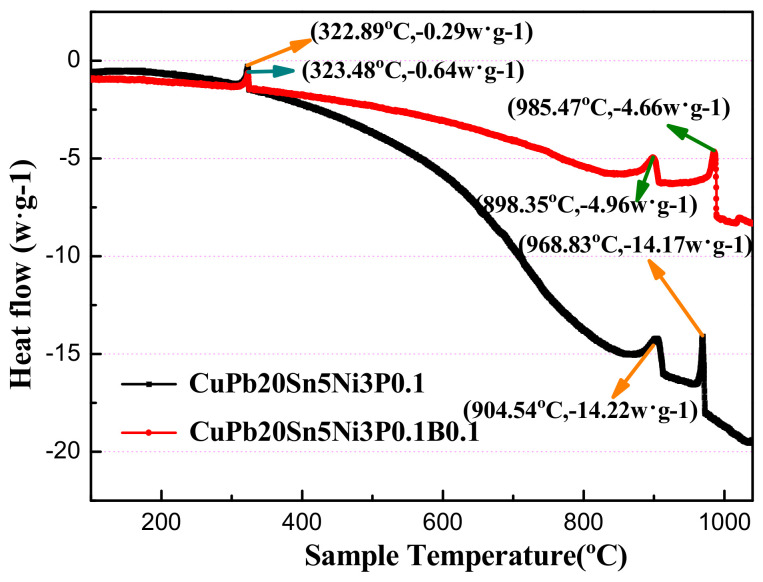
DSC of the ZCuPb20Sn5P0.1-xB alloy.

**Figure 7 materials-14-07806-f007:**
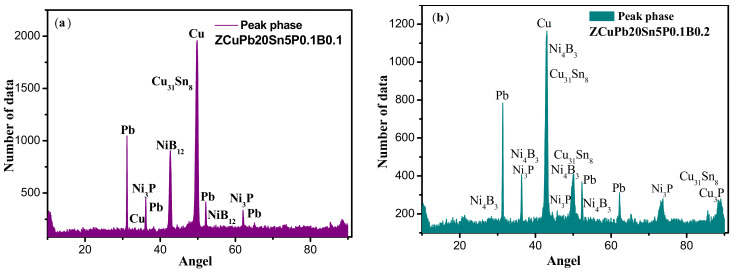
XRD analysis of the ZCuPb20Sn5-xB alloy. (**a**) ZCuPb20Sn5-0.1B; (**b**) ZCuPb20Sn5-0.2B.

**Figure 8 materials-14-07806-f008:**
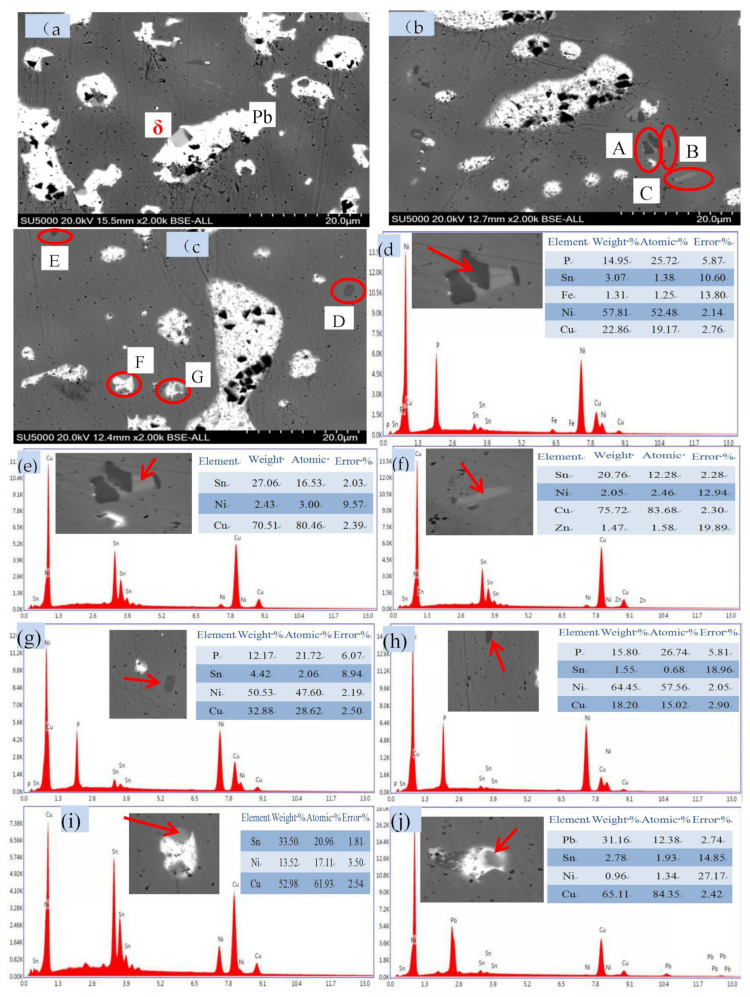
The phase microstructure of the ZCuPb20Sn5-0.1P-yB alloy. (**a**) SEM of alloy with 0 wt.% B; (**b**) SEM of alloy with 0.01 wt.% B; (**c**) SEM of alloy with 0.05 wt.% B; (**d**) EDS of A; (**e**) EDS of B; (**f**) EDS of C; (**g**) EDS of D; (**h**) EDS of E; (**i**) EDS of F; (**j**) EDS of G.

**Figure 9 materials-14-07806-f009:**
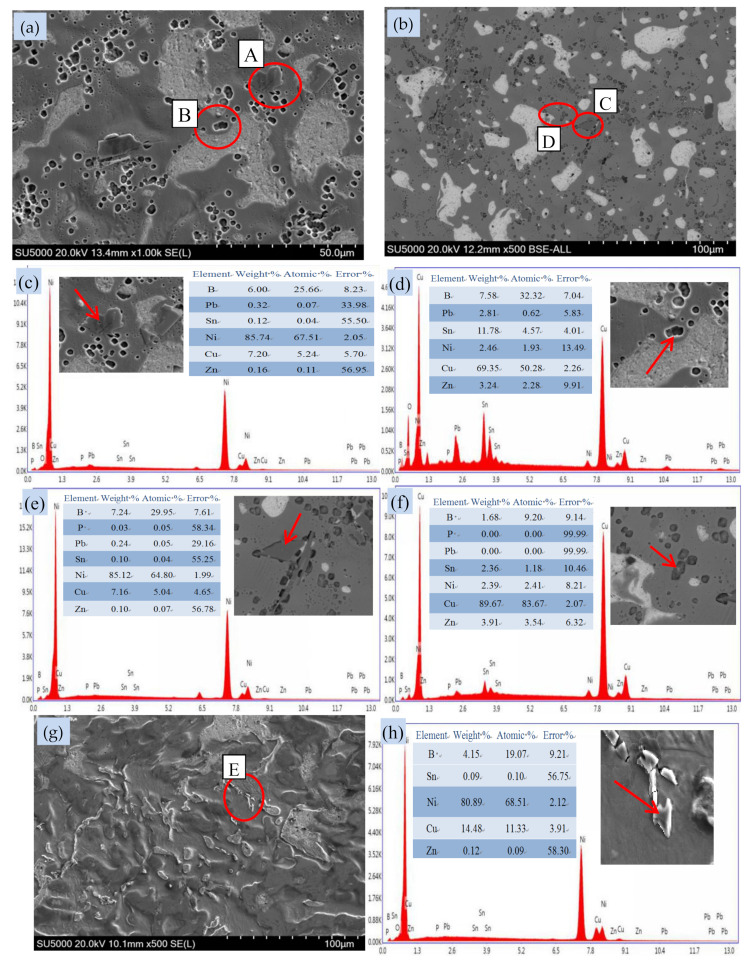
Microstructure of alloy ZCuPb20Sn5Ni3-0.1P-yB phase. (**a**) SEM of alloy with 0.1 wt.% B; (**b**) SEM of alloy with 0.2 wt.% B; (**c**) EDS of A; (**d**) EDS of B; (**e**) EDS of C; (**f**) EDS of D; (**g**) SEM of alloy with 0.2 wt.% B; (**h**) EDS of E.

**Figure 10 materials-14-07806-f010:**
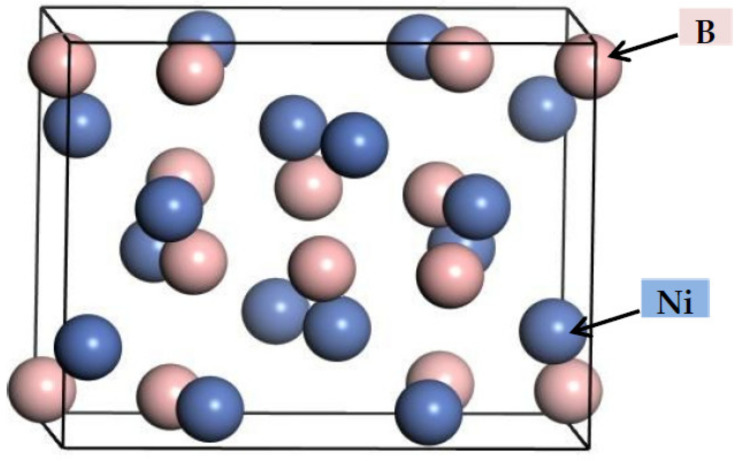
Crystal structure of Ni_4_B_3_ phases.

**Figure 11 materials-14-07806-f011:**
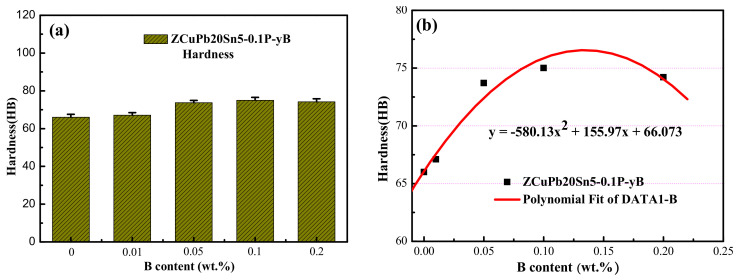
Hardness of ZCuPb20Sn5 alloy with different element contents. (**a**) Hardness of ZCuPb20Sn5-0.1P-yB; (**b**) ZCuPb20Sn5 alloy hardness fitting curve.

**Figure 12 materials-14-07806-f012:**
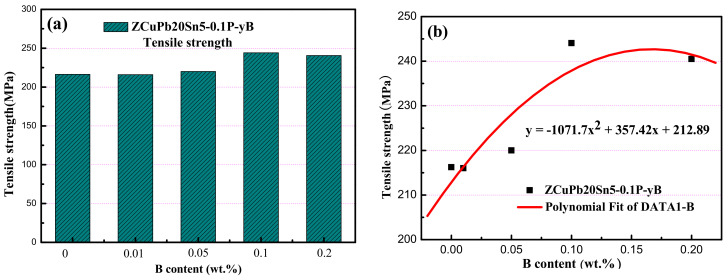
Tensile strength of ZCuPb20Sn5 alloy with different element contents. (**a**) tensile strength ofZCuPb20Sn5-0.1P-yB; (**b**) Fitting curve.

**Figure 13 materials-14-07806-f013:**
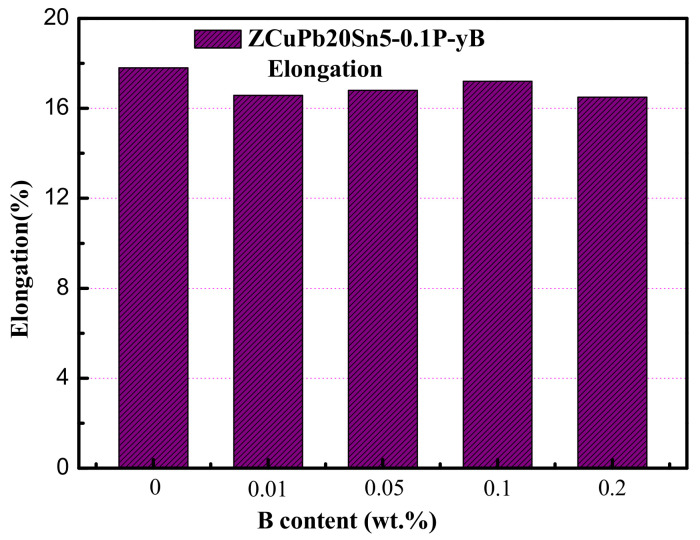
Elongation of ZCuPb20Sn5 alloy with different B element contents.

**Figure 14 materials-14-07806-f014:**
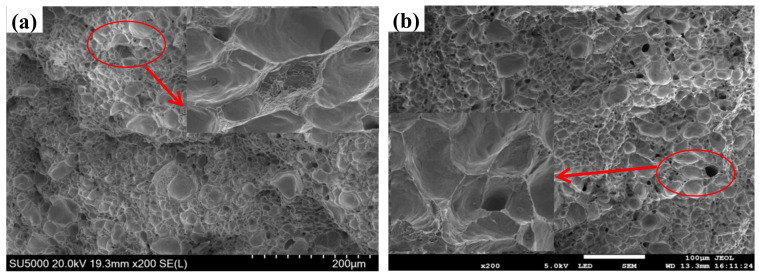
Tensile fractographs of ZCuPb20Sn5-yB alloy. (**a**) B 0.1; (**b**) B 0.2.

**Table 1 materials-14-07806-t001:** Data table of average diameter of lead particles after adding different elements to lead–tin bronze.

Element Content (wt.%)	Average Diameter of Lead Particles (μm)
B:0.00	30.0
B:0.01	28.8
B:0.05	26.8
B:0.10	24.8
B:0.20	28.6

**Table 2 materials-14-07806-t002:** Crystal structure parameters of Ni_4_B_3_ phases.

Phase	Group (No.)	Crystal Structure	Lattice Constant (Å)			Densityg/cm^3^	Volume cm^3^	Enthalpy of Formation eV/Atom	Binding Energy eV/Atom
			a	b	c				
Ni_4_B_3_	I-4	Triclinic system	6.392	4.913	7.868	7.348	241.6	−0.599	−6.247
Ni_3_P	I-4	Quartet	8.966	8.966	4.384	7.808	352.4	−0.510	−5.909
Cu_3_P	P6_3_CM	Six parties	6.971	6.971	7.183	7.305	302.3	−0.041	−4.158

**Table 3 materials-14-07806-t003:** Ni_4_B_3_ elastic constant table.

C11	C12	C13	C14	C15	C16	C22	C23
449.951	148.690	165.651	−0.075	−6.294	2.496	346.940	130.630
C24	C25	C26	C33	C34	C35	C36	C44
0.925	−49.817	0.292	342.235	1.309	−12.782	0.111	78.135
C45	C46	C55	C56	C66			
0.015	−15.545	83.919	−1.342	107.640			

**Table 4 materials-14-07806-t004:** Ni_4_B_3_ related modulus data sheet Unit: GPa.

	B_V_	B_R_	B_H_	G_V_	G_R_	G_H_	E	σ
Ni_3_P	204.35	204.31	204.33	63.22	60.90	62.06	169.07	0.36
Cu_3_P	114.43	113.31	113.87	45.51	44.27	44.89	119.03	0.33
Ni_4_B_3_	225.45	211.68	218.57	100.22	92.42	96.32	251.95	0.31

Remarks: Bulk modulus (B), shear modulus (G), Young’s modulus (E), Poisson’s ratio (σ).

## Data Availability

The data presented in this study are available upon request from the corresponding author.

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
