# Peer review of "Effects of B on the Structure and Properties of Lead-Tin Bronze Alloy and the Mechanism of Strengthening and Toughening"

_materials, 2021, doi:10.3390/ma14247806_

Round 1

Reviewer 1 Report

The manuscript investigates the influence of B on mechanical properties and microstructure. The manuscript contains many experimental results that are valuable to publish. The results are clearly presented. However, there are some suggestions for corrections and comments:

  1. Section 3.1.4., Figures 3-7 and 3-8. Please indicate for how many particles the chemical composition was measured and whether the values of the chemical composition in Figs. 3-7 and 3-8 are the average values of chemical composition.

  1. Figure 3-7. The results suggest that the chemical composition of the particles denoted as A, D, E is very similar. The particles A,D,E are probably the same particles. The particles B and C have also very similar chemical composition.

  1. Section 3.2.1., Figure 3-11 and Tab. 3-3. Some data given in the curve (Fig. 3-11 b) do not agree with the results given in Tab. 3-3. It means that the fitting equation is not correct.

  1. Section 3.2.2., Figure 3-12 and Tab. 3-3. Some data given in the curve (Fig. 3-12 b) do not agree with the results given in Tab. 3-3. Fitting is not correct (is not optimal). The curve estimates a max. HB around 0.12 wt.% B and then a significant decrease of HB at 0.22 wt.%B. However, this is not supported by results. Are the average values of HB shown in Fig. 3-12?

Author Response

For the teacher’s question, it has been revised, see attachment

Reviewer 2 Report

In this study, the effect of B on the structure and mechanical properties of lead-tin bronze alloy was investigated. Some revisions are suggested before publication.

1. Change figures and tables from 3-1 ~ 3-14 to 1~14.  

2. Wouldn't the fraction have more influence than the size of the lead particles in Figure 3-1 and 3-2? In each microstructure, the fraction of lead particles differs significantly. Do you have any data on the fraction of lead particles? If the size of the lead particles is more influential than the fraction, explain this in the text and add the references.

3. Why does the size of the lead particles increase in Figure 3-2(e)?  It seems that the fraction of fine lead particles can be increased, but what is the driving force that makes the size of the lead particles larger? 

4. It seems that the order of XRD analysis in 3.1.4 (2) should come before SEM/EDS analysis in 3.1.4 (1). When explaining the correlation between heat flow/sample temperature in 3.1.3, it is already explained in relation to XRD data. 

5. It would be better to describe the hardness properties first in the mechanical properties part, and then describe the tensile properties and the fracture surface.

Author Response

The question raised by the teacher has been revised, see attachment

Reviewer 3 Report

Major Comments 

  1. Exothermic peaks of ZCuPb20Sn5B0.1 Alloy were found from Heat Flow vs Sample Temperature. However, there is not sufficient information how Heat Flow was determined.
  2. Material properties like Bulk modulus, shear modulus, Poisson’s ratio was used to determine the hardness and ductile phase, it should be useful to describe how these properties are measured

Minor Comments

  1. First line of Abstract, thermai should be replaced by thermal.
  2. Third line of Abstract, full stop is missing before a new sentence beginning by Under.
  3. 4th line of Abstract, there should be comma before high speed and again after high strength and add few more commas in 5th line.
  4. 9th line of Abstract, there is no need of a attached with of.
  5. 14th line of Abstract, sentence, “The change … block” should be corrected to make a sense.
  6. 17th line of Abstract, replace colon by full stop.
  7. 26th line of Abstract, sentence starting with The main …, should better be reduced into two proper sentences.
  8. First sentence of Introduction is not clear. Need to be re-written.
  9. 5th line of Introdction starting with “At the same”and ending by “to” needs to be changed to make sense.
  10. 9th line of Introduction, ~ means nearly equal. This may be replaced by -.
  11. First line of 2nd para of Introduction, what research, mention here.
  12. Third line of 3rd para of Introduction, remove solid solubility and remove a full stop out of two.
  13. Last sentence of last para of Introduction is a lengthy sentence. Break into two sentences with correct English.
  14. Similar corrections in the rest of the manuscript need to be made.

Author Response

Thanks for the questions from the experts

All content in Minor Comments has been modified

Round 2

Reviewer 1 Report

The manuscript can be published in the present form.

Reviewer 2 Report

The author answered most of my comments. Therefore, I can accept this manuscript  now. Thanks.

Reviewer 3 Report

The authors of the present manuscript have provided us with the required answers for our questions and therefore I recommend this article for publication in its present form.